# Structural Heterogeneity of the Rabies Virus Virion

**DOI:** 10.3390/v16091447

**Published:** 2024-09-11

**Authors:** Xiaoying Cai, Kang Zhou, Ana Lucia Alvarez-Cabrera, Zhu Si, Hui Wang, Yao He, Cally Li, Z. Hong Zhou

**Affiliations:** 1Department of Microbiology, Immunology and Molecular Genetics, University of California, Los Angeles (UCLA), Los Angeles, CA 90095-1489, USA; xycai@g.ucla.edu (X.C.); anlualca@yahoo.com (A.L.A.-C.); siz@sustech.edu.cn (Z.S.); logicvay2010@g.ucla.edu (H.W.); yaohe@ucla.edu (Y.H.); 2The California NanoSystems Institute (CNSI), University of California, Los Angeles (UCLA), Los Angeles, CA 90095-7364, USA; zhoukdz@g.ucla.edu (K.Z.); cally40120@gmail.com (C.L.); 3Alsion Montessori High School, 750 Witherly Ln., Fremont, CA 94539, USA

**Keywords:** rhabdoviruses, rabies virus, wild type, cryogenic electron tomography, cryogenic electron microscopy, flexibility, dynamics

## Abstract

Rabies virus (RABV) is among the first recognized viruses of public health concern and has historically contributed to the development of viral vaccines. Despite these significances, the three-dimensional structure of the RABV virion remains unknown due to the challenges in isolating structurally homogenous virion samples in sufficient quantities needed for structural investigation. Here, by combining the capabilities of cryogenic electron tomography (cryoET) and microscopy (cryoEM), we determined the three-dimensional structure of the wild-type RABV virion. Tomograms of RABV virions reveal a high level of structural heterogeneity among the bullet-shaped virion particles encompassing the glycoprotein (G) trimer-decorated envelope and the nucleocapsid composed of RNA, nucleoprotein (N), and matrix protein (M). The structure of the trunk region of the virion was determined by cryoEM helical reconstruction, revealing a one-start N-RNA helix bound by a single layer of M proteins at an N:M ratio of 1. The N-M interaction differs from that in fellow rhabdovirus vesicular stomatitis virus (VSV), which features two layers of M stabilizing the N-RNA helix at an M:N ratio of 2. These differences in both M-N stoichiometry and binding allow RABV to flex its N-RNA helix more freely and point to different mechanisms of viral assembly between these two bullet-shaped rhabdoviruses.

## 1. Introduction

In 1885, Louis Pasteur and his French colleagues demonstrated the first successful anti-rabies vaccine by injecting rabid rabbit spinal cord materials into nine-year-old Josephe Meister [1]. This historical event occurred approximately seven decades before the rabies virus (RABV) was visualized using electron microscopy [2]. RABV, classified in the genus *Lyssavirus* within the family *Rhabdoviridae*, shares a characteristic bullet shape with a close relative, the vesicular stomatitis virus (VSV). Evidence of RABV morphological variability has been reported for some strains of the virus [3,4]. The family *Rhabdoviridae* belongs to the order of negative-strand RNA viruses (NSRV), *Mononegavirales*. Other notable NSRVs include the medically relevant influenza virus [5], Ebola virus [6], and Marburg virus [6,7]. Like VSV, RABV has been recognized as a tool virus for beneficial applications—it can be modified to serve as an anti-cancer agent because it displays high selectivity in killing cancer cells while sparing normal cells [8,9] and can also be engineered for vaccine development [10]. Three-dimensional (3D) structural information of the proteins composing RABV is thus highly significant and has been sought after for decades [11,12,13,14,15]. However, how these proteins are organized and assembled within the bullet-shaped RABV virion remains unknown.

The RABV genome encodes for five proteins: nucleoprotein (N), phosphoprotein (P), matrix protein (M), glycoprotein (G), and polymerase (L). All rhabdoviruses have two major structural components: a helical or bullet-shaped ribonucleoprotein (RNP) core and a surrounding envelope. The genomic RNA is wrapped in the helical assemblies of N, which encapsulate the polymerase complex, P and L. M resides between the RNP helix and the envelope, studded by G homotrimers. Difficulties in culturing and isolating large quantities of RABV virions have hindered the progression of structural studies. While the structure of fellow rhabdovirus VSV has been resolved at atomic resolution [16,17,18], the only structural knowledge of RABV comes from a low-resolution cryogenic electron tomography (cryoET) reconstruction of a pseudotyped virus of RABV with the G removed [12]. As a result, the structure of the RABV virion and its systematic comparison with that of VSV have been lacking—a significant gap in knowledge considering RABV’s impacts in vaccine history, public health, and bioengineering potentials.

In this study, we combined data from cryoET and cryogenic electron microscopy (cryoEM), determined the 3D structure of the trunk part of this virus, and depicted the architectural organization of the entire bullet-shaped virion. We demonstrate that RABV has a single layer of M proteins that binds with the underlying N-genomic RNA ribonucleoprotein helix differently as compared to VSV, which has two layers of M securing the N bullet. The crucial differences in M-N stoichiometry and binding allow for an increased flexibility in the N-RNA helix, thereby resulting in the structural heterogeneity of RABV.

## 2. Materials and Methods

### 2.1. Virus Strain and Cell Line

The CVS-27 RABV strain (VR-321; ATCC, Manassas, VA, USA) was used in this study. It was propagated in a BSL-2 facility under biosafety approval from the University of California, Los Angeles, with the approval code BUA-2015-742-011-CR. Neuro-2A (N2a) cells (CCL-131; ATCC, Manassas, VA, USA) served as hosts for virus propagation and were maintained in culture media DMEM/Hams F12 (Cat No. 10-090-cv; CORNING, New York, NY, USA) and Opti-MEM (Cat No. 31985070; GIBCO, Carlsbad, CA, USA) at a 1:1 ratio, supplemented with 2% (*v*/*v*) heat-inactivated fetal bovine serum (FBS) (Cat No. 30-2003; ATCC, Manassas, VA, USA), penicillin (50 U/mL), and streptomycin (50 μg/mL) (Pen/Strep) (Cat No. 15140122; GIBCO, Carlsbad, CA, USA), at 37 °C in a humidified atmosphere of 5% carbon dioxide (CO_2_) and 95% oxygen (O_2_).

### 2.2. Virus Production and Purification

N2a cells at 60% confluence, grown in 18 × T175 (175 cm^2^ area) tissue culture flasks, were infected with RABV virus stock dispersed in 15 mL of culture media per flask and incubated for 1 h before adding fresh culture media without FBS. After 72 h post-infection, the virus-containing media was harvested and clarified by centrifugation at 10,000× *g* for 30 min in a Fiberlite F14-6 × 250y Fixed-Angle Rotor (Thermo Fisher Scientific, Waltham, MA, USA) at 4 °C. All subsequent sample purification steps were carried out at 4 °C. RABV virions were initially pelleted by ultracentrifugation at 75,000× *g* (SW 28 Ti Swinging-Bucket Rotor; Beckman Coulter, Brea, CA, USA) for 1 h, after pooling the clarified supernatant on top of a 10% iodixanol cushion prepared from a dilution of a 60% (*w*/*v*) aqueous stock solution of OptiPrep (Cat No. D1556; Sigma-Aldrich, St. Louis, MO, USA) with phosphate-buffered saline (PBS) of pH 7.4 (Cat No. 10010049; GIBCO, Carlsbad, CA, USA). The pellets were resuspended in PBS of pH 7.4, pulled together, and subsequently loaded on top of a continuous 5–50% iodixanol density gradient. After ultracentrifugation at 75,000× *g* (SW 41 Ti Swinging-Bucket Rotor; Beckman Coulter, Brea, CA, USA) for 1 h, fractions of approximately 0.5 mL each were carefully collected from the top of the tube. Each fraction was slowly diluted in PBS of pH 7.4 and subjected to ultracentrifugation at 75,000× *g* (SW 41 Ti Swinging-Bucket Rotor; Beckman Coulter, Brea, CA, USA) for 1 h. The individual pellets were dispersed in PBS of pH 7.4, and each suspension was evaluated by negative staining electron microscopy to identify fractions containing RABV virions.

### 2.3. CryoEM Sample Preparation and Movies Acquisition

For single-particle cryoEM, an aliquot of 2.5 μL of purified RABV virions was applied onto a glow-discharged lacey carbon grid with a supporting ultrathin carbon film (Ted Pella, Redding, CA, USA). Grids were then blotted and plunge-frozen in liquid ethane with a manual plunger.

CryoEM grids were loaded into a Titan Krios 300 kV electron microscope (Thermo Fisher Scientific, Waltham, MA, USA) equipped with a Gatan imaging filter (GIF) Quantum LS and a Gatan K3 direct electron detector. Movies were acquired with SerialEM v3.9 [19] in super-resolution mode at a nominal magnification of 81,000×, yielding a calibrated pixel size of 0.55 Å at the specimen level. The GIF slit width was set to 20 eV. For each movie, a total number of 40 frames were acquired, with a total cumulative dose of ~50 e^−^/Å^2^. The defocus range was −1.8 to −2.1 μm.

### 2.4. CryoET Sample Preparation and Tilt Series Acquisition

For cryoET, purified RABV virion sample was mixed with 10 nm fiducial gold beads at a ratio of 1:20 (*v*/*v*) before freezing. An aliquot of 2.5 μL of the mixed sample was applied onto a glow-discharged lacey carbon grid with a supporting ultrathin carbon film (Ted Pella, Redding, CA, USA) and plunge-frozen in a manual plunger.

Grids were loaded onto the same Titan Krios 300 kV electron microscope (Thermo Fisher Scientific, Waltham, MA, USA). CryoET tilt series were obtained with SerialEM v3.9, with a dose-symmetric tilt-scheme. The beam was aligned in nano-probe mode and the GIF slit width was set to 20 eV. Tilt series movies were recorded at a nominal magnification of 64,000× (calibrated pixel size of 0.69 Å), with a total cumulative dose of approximately 120 e^−^/Å^2^. The tilt range was from −60° to 60°, with a step size of 3°, and the defocus range was from −2.5 to −4.5 μm. Dose-fractionated frames were 2× binned (pixel size 1.38 Å) and aligned for drift correction with the graphics processing unit (GPU)-accelerated program MotionCor2 [20]. A total of 41 tilt series were generated, and they were used for the following tomogram reconstruction.

### 2.5. CryoET and Subtomogram Averaging (STA)

CryoET data processing was performed with TomoNet [21], IsoNet [22], and Relion4 [23]. The defocus value of each tilt image was estimated using CTFFIND4 [24] integrated in TomoNet. Fiducial-based tilt series alignment and tomographic reconstruction were performed with IMOD [25] integrated in TomoNet. Fiducial gold beads were manually picked and automatically tracked. The fiducial model was corrected when the automatic tracking failed. The final alignment was computed without solving for any distortions. Tomograms were reconstructed with the SIRT-like filter option in IMOD under 4-binned pixel size. Thirty-four tomograms containing thirty-five RABV virions were reconstructed. For better visualization, the above tomograms were denoised and missing-wedge corrected using IsoNet. The denoised tomograms were only used for displaying in the figures and for particle picking for STA. All measurements of the length and diameter of virions were obtained along their orthogonal axes.

The subtomogram-averaged density map of M-N was determined from ten RABV virions appearing with the typical bullet shape. The initial reference map of M-N was generated from ~500 manually picked particle segments, chosen based on the best visibility of M and N proteins. Then, 7534 particle segments containing M and N proteins were picked using the “Auto Expand” function in TomoNet, with rough coordinates and orientations determined. Particle segments with a cross-correlation coefficient lower than 0.25 were removed during the “Auto Expand” process, and the particles’ information was exported into a STAR format file as the input of Relion 3D refinements. After re-extracting particles with a box size of 96^3^ under 3-binned pixel size, a subtomogram-averaged density map was determined through two rounds of 3D refinement using Relion4, with no symmetry applied in the reconstruction process. After an additional round of 3D classification, 2300 particles from the best class were selected and refined to 18.7 Å resolution. The resolution reported above is based on the “gold standard” [26] refinement procedures and the 0.143 Fourier shell correlation (FSC) criterion.

### 2.6. Single-Particle CryoEM Data Processing

Frames in each movie were aligned for drift correction with the graphics processing unit (GPU)-accelerated program MotionCor2 [20]. The first and last frames were discarded during drift correction. Two averaged micrographs, one with dose weighting and the other one without dose weighting, were generated for each movie after drift correction. The averaged micrographs were 2× binned to yield a pixel size of 1.1 Å. The micrographs without dose weighting were used for CTF estimation and particle picking, while those with dose weighting were used for particle extraction and in-depth processing.

The data processing workflow for RABV single-particle cryoEM dataset is summarized in Appendix A. The CTF estimation of each micrograph was performed by CTFFIND4 [24]. From a total of 1622 micrographs, 307 RABV virions (start–end coordinates pair) were selected manually using Relion3 [27]. With helical extraction function in Relion, each virion was sub-divided into overlapping segments with dimensions of 1120 × 1120 square pixels and a 7.8% inter-box distance of 96 Å. A total of 3906 particles (segments) were extracted and 2× binned to 560 × 560 square pixels (pixel size: 2.2 Å) to speed up further data processing. These particles were then subjected to three rounds of reference-free 2D classifications. Classes with unfavorable particles (i.e., classes with fuzzy or uninterpretable features) and those containing either the virion tip region or the bottom region were discarded, yielding a total of 1693 selected particles. These particles were subjected to 3D classification with a single class (K = 1) and a featureless hollow cylinder created by EMAN2 [28] as initial model. Using one helical symmetry (helical twist: −7.50°, helical rise: 1.20 Å) roughly measured from cryoET dataset, the cylinder was refined to a helical model in which fuzzy helical features and two membranes were observed. This helical model was then used as initial model to classify the 1693 particles again in 3D classification. Using helical symmetry search range (helical twist: −5.90°–−7.90°, helical rise: 1.00 Å–1.40 Å), particles were classified into 6 classes (Appendix A). A total of 560 particles (segments) in the best class were subjected to further 3D classification with 3 classes (K = 3). Then, 44 particles (segments) in the best class (helical twist: −6.89°, helical rise: 1.37 Å) were selected for further processing.

Owing to the flexibility of the virion helices, the resolution of helical segments was limited to ~20 Å. To accommodate for the flexibility and improve resolution, small sub-particles were re-extracted from the helical segments. In detail, using the command “relion_particle_symmetry_expand”, 44 segments were expanded to 3080 segments where each segment was expanded to 70 patches with 1 sampling in each asymmetric unit. By shifting the center in X-coordinate for 120 pixels (2.2 Å/pixel in expanded segments, 264 Å), 3080 sub-particles with 192 × 192 square pixels (1.1 Å/pixel in original micrographs) were re-extracted and 2× binned to 96 × 96 square pixels (2.2 Å/pixel). Using a RELION reconstructed map as an initial model, 3080 sub-particles were subjected to a 3D local auto-refinement step and a post-processing step, yielding a final map of the trunk at 11.1 Å resolution (Appendix A).

## 3. Results

### 3.1. Structural Flexibility of RABV Virions

To directly visualize the 3D organization of wild-type RABV, we collected cryoET tilt series from RABV samples purified through continuous iodixanol density gradient ultracentrifugation. This process was designed to minimize the presence of defective interfering particles with incomplete genomic RNA [29] in RABV virions. Reconstructed tomograms were denoised and missing-wedge corrected with IsoNet [22] to enhance observation of virion morphology.

RABV virions typically have an enveloped, bullet-shaped appearance, a feature commonly seen in the rhabdoviruses family [30]. However, 74% of the virions in the reconstructed tomograms displayed structural variations deviating from the expected typical bullet shape (Appendix A). These structural variations were identified by comparing the diameters of individual virions. For typical bullet-shaped RABV virions, cross-section views at different heights of the virion trunk appeared roughly circular with consistent diameter (Figure 1A), indicating that the trunks of the virions were cylindrical. The observed structural variations of the analyzed RABV virions were categorized into two types. The trunk of the first type appears as an elliptical cylinder. These virions reveal an oval shape in the cross-section views at different heights of the trunk, with comparable minor diameters and similar major diameters (Figure 1B). Since the minor diameters approximate the thickness of the vitreous ice (Figure 1B), this variation likely occurred during the cryoEM specimen freezing process, where the virions were flattened under the blotting force. The second type of structural variation exhibits a gradual alteration in virion trunk morphology. The cross-section views of these virions at different heights of the trunk show a progressive change in shape, with diameters varying by approximately 10 nm or more (Figure 1C). Overall, RABV has a flexible architecture and is prone to deformation under mechanical stress, a characteristic not observed in VSV structures.

### 3.2. Variable Pitch and Larger Diameter of RABV Compared to VSV

To investigate the underlying factors contributing to RABV’s structural flexibility and its susceptibility to deformation, we conducted comparative analyses of its diameter, length, and the pitch of helical RNP in relation to those of VSV. An average diameter of 85 nm and an average length of 178 nm were derived from thirty-five RABV virions in reconstructed tomograms. This diameter closely approximates the previously reported average of 86 nm for the G gene-deficient RABV particles [12]. While the diameter range (Figure 2A,C) aligns with that of G gene-deficient RABV, our study revealed a broader length distribution and a shorter average length (Figure 2A,B) compared to G gene-deficient RABV, which has an average length of 198 nm, with a range of 183 to 222 nm [12]. The presence of particles assembled with defective viral genomes likely results in shorter virions, which may explain the broader length distribution and shorter average length found in our study. Although RABV and VSV share similar left-handed helical RNP [11,16,17,31,32], the helical parameters of the two virions are distinctly different. A comparison of the diameters reveals that RABV exhibits a diameter approximately 1.2 to 1.3 times larger than that of VSV (Figure 2D,E), implying a larger number of asymmetric units per turn. In contrast to the constant pitch observed in the VSV RNP helix [16,17,32], the pitch of the RABV RNP varies between 5.7 and 7.1 nm across 35 virions (Figure 2F). Additionally, the pitch of the helical RNP in the RABV trunk is longer than that in VSV, with an average pitch of 6.3 nm measured from 35 virions, surpassing the constant VSV pitch of around 5 nm [16,17,32]. The variability in pitch suggests a potential for structural variability in RABV.

Structural variation of the tip was captured in our reconstructed tomograms of RABV virions, underscoring a distinctive feature of RABV compared to VSV. A conical tip of RABV is composed of eight turns of a conical spiral (Figure 2G), as evident in the VSV conical tip [17,18]. The end of the tip is defined by the horizontal angles of these turns, measured from the long axis of the M-N proteins along the horizontal plane of the virion. These angles decrease progressively from turns 1 to 8 and then remain constant at the start of the trunk (Appendix A). Additionally, dome-like and flat-topped tips were detected in RABV virions, composed of five turns and three turns of spirals, respectively (Figure 2G and Appendix A). This variation in the number of spiral turns at the tip implies a flexible conic helix architecture in RABV tips, which may be related to the absence of a fixed helical pitch. Beyond these structural differences at the tip, variations in horizontal angles within the RABV trunk were observed. The horizontal angle within the trunk of a single virion remains constant; however, these angles vary across virions with the three aforementioned types of tips and are not related to the number of spiral turns at the tip (Appendix A). This variability in the horizontal angle within the trunk across different virion individuals further highlights the diverse architecture of RABV virions.

An in situ structure of M-N, determined by STA, was placed back into the tomograms to confirm the variable pitch in the RABV trunk. Subtomograms containing M and N proteins were extracted from ten RABV virions appealing with the typical bullet shape (Appendix A) and then aligned and averaged. The M-N subtomogram average density in one turn of the RNP helix was cropped (Appendix A) and eventually placed back into the original tomograms based on the coordinates and orientations of the aligned subtomograms. This process provides an overall in situ distribution of the M-N asymmetric unit (Figure 2I) and allows for a more accurate placement of M-N within the helical turns of the RNP. Such placement facilitates the direct verification of the pitches measured from the reconstructed tomograms. The average pitches for each of the ten virions, measured from the placed-back M-N subtomograms, vary consistently with the observations in reconstructed tomograms, sharing a similar pitch median and pitch distribution (Figure 2F). A pitch of 6.3 nm was observed in the subtomogram average of M-N (Appendix A), closely aligning with the average pitches measured from both the reconstructed tomograms and the placed-back M-N subtomograms (Figure 2F). Additionally, the pitch variation, even within a single virion, was observed in the placed-back M-N subtomograms (Figure 2J), further implying that the irregular assembly of the RNP helix may compromise the structural stability of RABV virions. An averaged density map with enhanced features of the M-N subunits was obtained after an additional round of 3D refinement, followed by 3D classification of the previously aligned subtomograms using the skip-align option (Figure 2H). This structure was determined at a resolution of 18.7 Å (Appendix A). The adjacent two M subunits in one helical turn and the two main domains of the N protein in this averaged density map are ambiguously observed. The neighboring N units are 3.8 nm apart, which is slightly larger than the 3.5 nm separation in the N units of G gene-deficient RABV [12].

By integrating the measurements from the reconstructed tomograms and the 3D distribution of the asymmetric unit with the placed-back subtomograms, we discovered that RABV RNP helix has a larger diameter and a variable pitch compared to VSV RNP. The increased diameter suggests that the torsion—the force required to support the helical shape of the RABV RNP—is smaller than that of VSV [33,34]. A constant torsion is crucial for maintaining a stable helical structure [33,34], but this variability in pitch indicates that the torsion of the RNP helix is not constant. The smaller and variable torsion of the RNP helix suggests that RABV has a more flexible and deformable architecture, in contrast to the more rigid structure of VSV.

### 3.3. In Situ Structures of M-N by Helical Sub-Particle Reconstruction

To obtain a higher-resolution structure of RABV, single-particle cryoEM micrographs of RABV virions were recorded, followed by helical sub-particle reconstruction (see Section 2 for details). Using the variable helical parameters measured from the cryoET reconstructions, we were able to obtain 3D classes from the cryoEM dataset (Appendix A). By focusing on the largest number of particles and performing helical sub-particle reconstruction, we obtained a 3D structure of the RABV M-RNP at a 11.1 Å resolution (Appendix A). Although the resolution limited our ability to map out the atomic details of the RABV trunk, we were still able to observe the organization and dock pattern of N and M in RABV (Figure 3 and Appendix A). The cryoEM density reveals the 1:1 ratio between N and M (Figure 3A and Appendix A), which is notably different from the ratios observed in VSV [16,18].

The crystal structure of the RABV N monomer was previously reported [11,35], while the M structure remains unknown. We used AlphaFold2 [36] to generate a model of an RABV M monomer. This allowed us to fit the models of N and M into a region of the cryoEM sub-particle reconstruction, which includes 8 N and 8 M (Figure 3B). Each asymmetric unit of RABV trunk contains 1 N and 1 M. Within the same helical turn, the lateral interactions between adjacent N subunits provide the main constraints, which is similar to those in VSV (Figure 3B,G,H). The single-strand RNA chain passes through N and enhances the lateral stability of the RABV trunk. Each M subunit primarily interacts with one N subunit in the same asymmetric unit (Figure 3C,D). While adjacent M proteins within the same turn may interact, M proteins from different turns are too distant to interact (Figure 3B).

Structure comparisons between RABV and VSV [16] reveal that N structures in both viruses are highly conserved, although there is a slight difference in the tilt angles of N between RABV and VSV (Figure 3C,E). In addition to lacking a second M layer in RABV, the organization in the RABV trunk has several other significant differences. First, the single M subunit in RABV is positioned along the extended radial line of N and mainly interacts with one N, limiting N-M interactions to within the asymmetric unit (Figure 3B–D). On the contrary, in VSV, the inner M interacts with two adjacent N subunits within the same turn, as well as with N subunits from neighboring turns, which reinforces lateral and vertical constraints (Figure 3G,H). Second, in VSV, the inner M has a long N-terminal tail that interacts with another inner M from the neighboring turn [16,18]. This unique interaction is thought to determine the distance between two turns in VSV, maintaining a nearly constant “turn rise”. However, the orientation of M in RABV has significant discrepancies compared to that of M in VSV (Figure 3D,F), suggesting that the N-terminal tail interaction may be disrupted in RABV. Consequently, we concluded that the distances between turns in RABV greatly varies, which is likely attributed to the absence of vertical M-M interactions.

**Figure 3 viruses-16-01447-f003:**
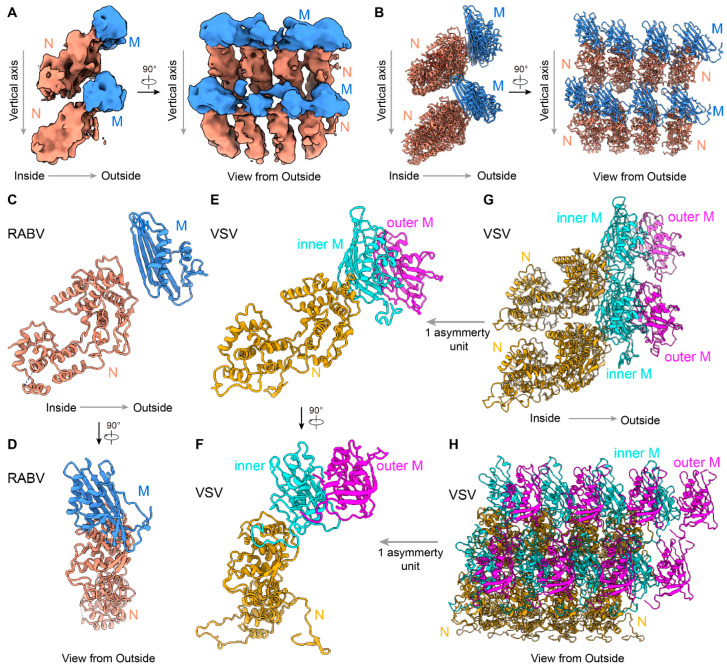
Sub-particle reconstruction of RABV trunk by cryoEM. (**A**) CryoEM density maps of partial RABV nucleocapsid shown in two orthogonal views. An 8 Å low-pass filter was applied to the cryoEM map for better visualization. Only intact N and intact M are colored for clarity, other densities are hidden. N subunits are colored in orange, and M are colored in blue. (**B**) Two orthogonal views of the fitted N protein models (PDB: 2GTT) [11] and AlphaFold2-predicted M protein models [36] corresponding to the cryoEM density maps in (**A**). (**C**,**D**) Two orthogonal views of the one asymmetry unit from RABV. RNA was hidden for clarity. (**E**,**F**) Two orthogonal views of the one asymmetry unit from VSV. N subunits are colored in yellow, inner M in cyan, and outer M in magenta. (**G**,**H**) Two orthogonal views of 8-asymmetry unit from VSV. A cross-sectional view is provided in (**G**) to more clearly display the models’ details.

Overall, compared to the VSV trunk, RABV nucleocapsid lacks many interactions in lateral, vertical, and radial directions, leading to significant instability and heterogeneity of the RABV nucleocapsid.

### 3.4. Both Prefusion and Postfusion G Trimers Captured on the Surface of RABV Virions

Though some recombinant G protein structures have been reported [14,15], knowledge regarding their in situ structure remains limited. In the reconstructed tomograms, we captured G proteins predominantly as trimers on the membrane envelope of RABV, existing in both prefusion and postfusion conformations (Figure 4A–D). The structure of the G trimers in prefusion conformation is shorter and wider than the structure in postfusion conformation (Figure 4A–C), similar to the observations of G trimers in the prefusion and postfusion conformations from VSV [16]. The average height and width of the prefusion G trimers are 10 nm and 9 nm, respectively (Figure 4A,B), matching the cryoEM structure of the recombinant RABV G trimer in prefusion conformation [15]. The atomic model corresponding to this structure can also be properly fitted into the density of the prefusion G trimer segmented from a reconstructed tomogram (Figure 4E). The postfusion G trimers, with an average height of 14.5 nm and an average width of 7 nm, were demonstrated in the reconstructed tomograms, presenting a more extended conformation than the prefusion conformation (Figure 4A,C). Since the virions were not treated with an acidic pH, the conversion from prefusion to postfusion conformation may have been induced by the density gradient ultracentrifugation process, a similar occurrence reported for VSV virions [16].

## 4. Discussion

By combining cryoET and cryoEM approaches, we managed to overcome the limitation of sample scarcity and heterogeneity to obtain the first structural information to describe the structural organization of the wild-type RABV virion. With parameters estimated from cryoET analysis, we have determined the 3D structure of the trunk at 11.1 Å resolution by helical reconstruction. By integrating the sub-nanometer resolution cryoEM structure of the trunk, along with reported and predicted models of individual proteins and their locations from cryoET tomogram of the virions, we have been able to depict the architecture of the trunk of RABV virion (Figure 5).

The structure of RABV virion reveals two major differences from that of fellow NSRV VSV: M:N stoichiometry and M-N binding orientation in the virions (Figure 3). The lack of a second M layer (the outer M layer in VSV) allows RABV to flex its N-RNA helix more freely than that of VSV, resulting heterogeneity or variability in RABV virion structures extensively documented here (Figure 1 and Figure 2). There is a possibility that different strains of rabies virus, or even virions of the same strain that are obtained after different number of rounds of cell passage, could have a varying degree of structural heterogeneity. While the biological relevance of such structural variability is hard to pinpoint at this time, it might enable RABV to overcome selective pressure as recently suggested by a systematic study based on pleomorphic influenza A virus, another negative strand RNA virus [37]. Furthermore, the correlation between the M1:N stoichiometry ratio and pleomorphism in influenza A viruses [38] suggests that the M:N ratio discrepancy contributes to the structural heterogeneity between RABV and VSV.

The distinct M organizations in RABV and VSV virions point to possible mechanistic differences in the virion assembly and budding of the progeny virus from host cell membranes (Figure 6). The current understanding of VSV assembly is that the inner M layer attaches to the N-RNA helix within the cytoplasm immediately after the N-RNA helix is assembled and the polymerase complexes are packaged [16]. Due to the interactions between M proteins and the cytoplasmic tail of the G trimers, the unbound M proteins accumulate on the underside of the cytoplasmic membrane at the budding site. Meanwhile, the hexagonal array of G trimers forms on the outer surface of the membrane [16]. The M-N-RNA nucleocapsid acquires the second layer of M proteins and creates a bud on the cytoplasmic membrane, resulting in the pinching off of a progeny virion and the release into the extracellular space (Figure 6). As for RABV, the exact location where M and N binding occurs remains unclear. It has been demonstrated that M proteins accumulate at cytoplasmic membranes [39] and interact specifically with G proteins [40]. Thus, M proteins might bind to the cytoplasmic tail of cellular membrane-anchored G trimers, as observed in VSV. Given that interactions between the inner M layer contribute to the fixed pitch of the VSV helical RNP [16], the variability of the pitch in the helical RNP of RABV virions suggests the absence of such interactions, which would otherwise stabilize the N-RNA helix pitch immediately after its formation in the cytoplasm. A possible scenario is that the binding of N-RNA and M proteins occurs at cytoplasmic membranes during RABV virion budding (Figure 6). The bud formation of the RABV N-RNA nucleocapsid might depend on their affinity to the M with N and might not be as frequent as that of the VSV, leading to low efficiency of RABV virion production. Future cellular cryoET of actively assembling RABV within a host cell should clarify these possibilities of RABV genome packaging and virion assembly.

## Figures and Tables

**Figure 1 viruses-16-01447-f001:**
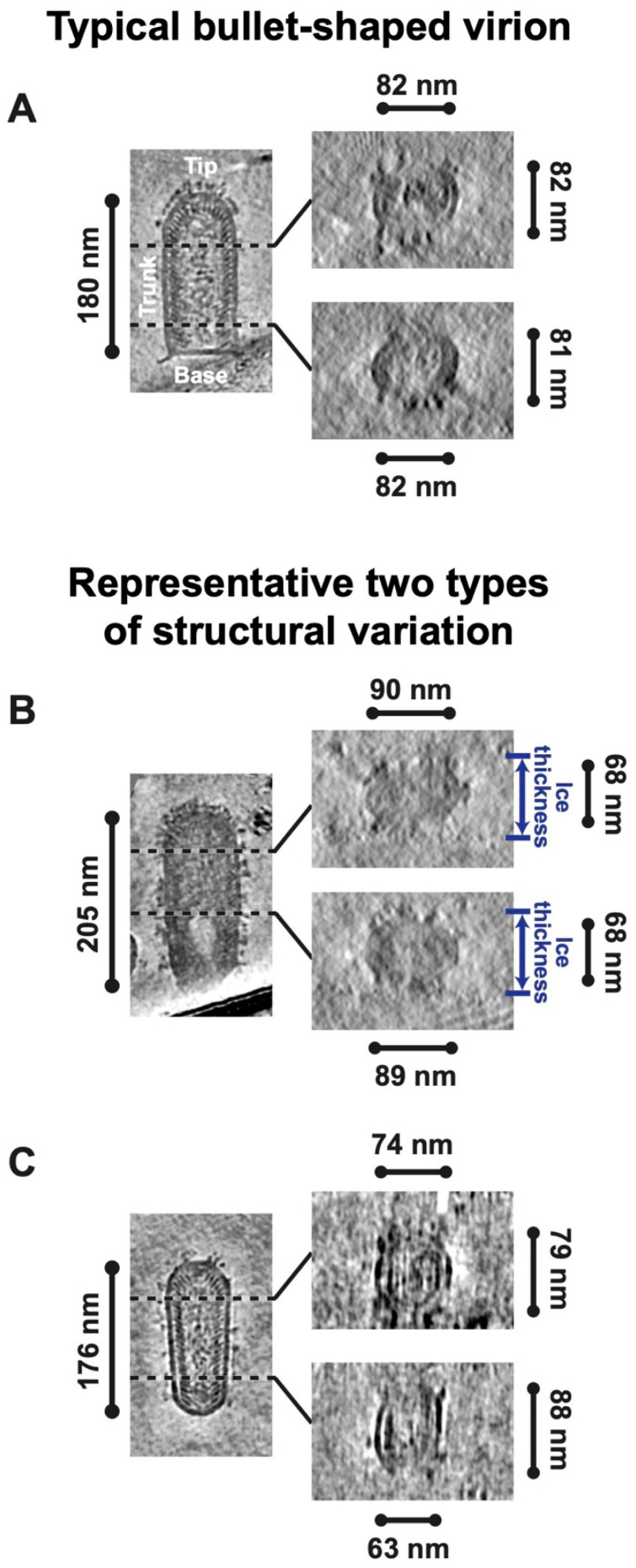
Typical bullet-shaped virion and its structural variations of rabies virus (RABV). (**A**) Depiction of 7 Å thick density slices from a reconstructed tomogram displaying a typical bullet-shaped RABV virion. The longitudinal central section of the virion is shown on the left, and the perpendicular sections at different heights along the virion trunk are presented on the right. (**B**,**C**) Depiction of 7 Å thick density slices from reconstructed tomograms displaying two types of structural variations of RABV virions: an overall flatten virion (**B**) and a virion with gradual alterations in trunk morphology (**C**). The longitudinal central sections of the virions are shown on the left, and the perpendicular sections at different heights along the virion trunk are presented on the right.

**Figure 2 viruses-16-01447-f002:**
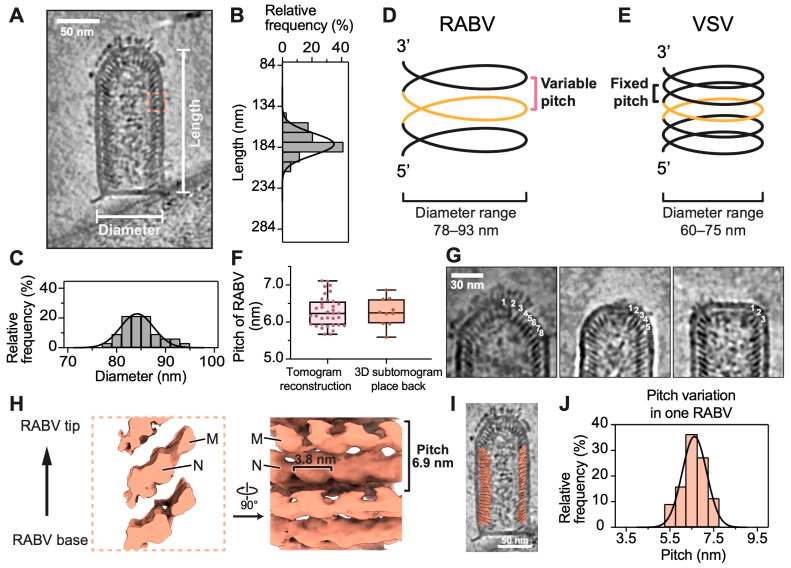
Architecture analysis of the helical RABV nucleocapsid by cryoET. (**A**) A slice through a reconstructed tomogram showing a section of a representative RABV virion. The scale bar corresponds to 50 nm. (**B**,**C**) The horizontal and vertical histograms show the distribution of measured length (**B**) and diameter (**C**), respectively, from thirty-five RABV virions. The solid black lines are normal distribution fits to the histograms with means and standard deviations of 178 ± 15 nm for the length and 85 ± 4 nm for the diameter. (**D**,**E**) RNP helix model of RABV and vesicular stomatitis virus (VSV). One turn of RNP helix is colored in yellow. (**F**) The box plots of the pitches measured from 35 reconstructed tomograms (left) and from placed-back M-N subtomograms in 10 reconstructed tomograms (right). Boxes and whiskers show interquartile range and maximum and minimum values of data; center lines in boxes represent the median. The means and standard deviations of the pitches measured from the reconstructed tomograms and the placed-back M-N subtomograms are both 6.3 ± 0.4 nm. (**G**) Three types of RABV tips are shown in slices from reconstructed tomograms in the order of conical, dome-like, and flat-topped tips. The turn layers of the helical RNP in the tips are annotated. The scale bar corresponds to 30 nm. (**H**) Side (left) and top views (right) of the subtomogram-averaged structure of RABV M-N. The dashed box in (**A**) shows a sample subtomogram extraction. The arrow on the left indicates the directionality of the virion. The pitch of RNP helix and the distance between subsequent units in one turn are indicated. (**I**) Placed-back M-N subtomograms in tomogram of (**A**). The place-back is guided by the coordinates and orientations of the aligned subtomogram in Appendix A. The scale bar corresponds to 50 nm. (**J**) The histogram displays the distribution of pitch measurements obtained from the placed-back M-N subtomograms within (**I**).

**Figure 4 viruses-16-01447-f004:**
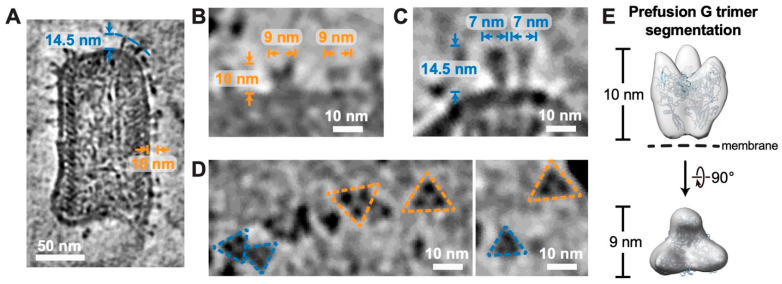
G trimers in both prefusion and postfusion conformations on RABV virion envelope. (**A**) A 7 Å thick density slice from a reconstructed tomogram showing a RABV virion with both prefusion and postfusion G trimers on viral envelope. The average height of 10 nm for glycoproteins indicates the prefusion conformation, whereas the average height of 14.5 nm corresponds to the postfusion conformation. The scale bar corresponds to 50 nm. (**B**,**C**) The side views of representative G trimers in prefusion (**B**) and postfusion conformations (**C**). Length and width of the G trimers are annotated. (**D**) The top views of G trimers. Trimers in prefusion conformation are framed by yellow dashed lines, while those in postfusion conformation are framed by blue dashed lines. The scale bar corresponds to 10 nm. (**E**) Prefusion G trimer segmented from a reconstructed tomogram. The atomic model of G trimer (PDB: 7U9G, ribbon diagrams colored in light blue) was docked into the segmented density map.

**Figure 5 viruses-16-01447-f005:**
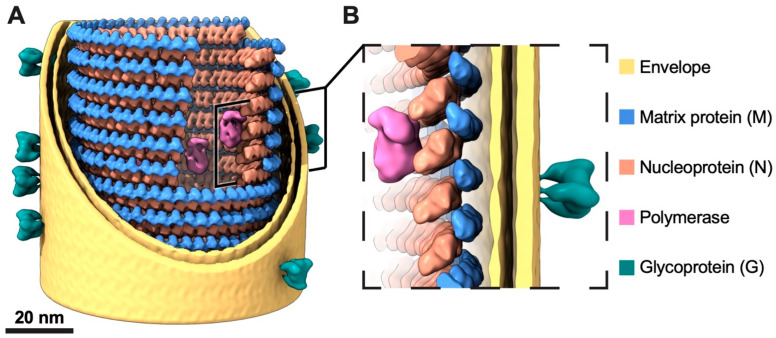
A model of the trunk of an RABV virion. (**A**) The trunk model of an RABV virion, with partial removal of the M and N proteins and the envelope membrane bilayer, enhances visual clarity and shows the interior structure. Two nested left-handed helices, consisting of an outer layer M (colored in blue) and an inner layer N (colored in orange), along with the envelope membrane bilayer (colored in yellow), were modeled based on the cryoEM density map of the trunk. The published cryoEM reconstructions of the RABV polymerase (EMD-20753, colored in magenta) and glycoprotein (EMD-26397, colored in dark cyan) were low-pass filtered to 20 Å and integrated into the model according to their positions within the virion density segmented from a reconstructed tomogram. The scale bar corresponds to 20 nm. (**B**) A zoomed-in view of a section of the RABV structural model.

**Figure 6 viruses-16-01447-f006:**
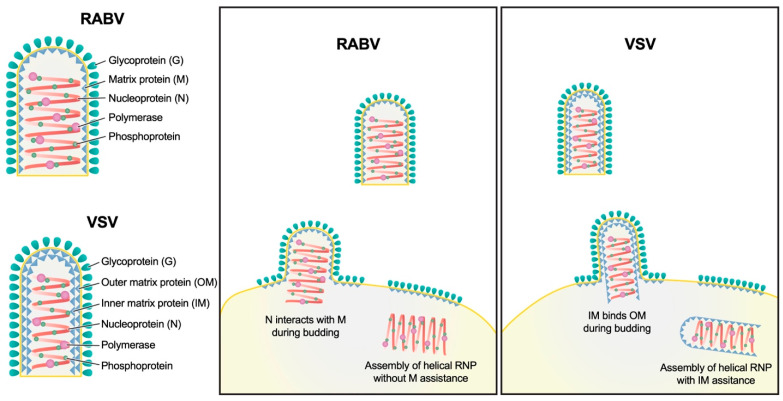
Schematic model illustrating the process of RABV assembly and budding and the differences with VSV. N of the helical RNP of VSV interact with the inner matrix protein (IM). Because the IMs in neighboring turns of the helix interact with each other, the pitch is fixed during the assembly of the VSV RNP helix. Subsequently, the outer matrix protein (OM), distributed on the cytoplasmic side of the envelope membrane lipid, binds to IM-RNP helix during virion budding. The helical RNP of RABV, on the contrary, assembles without assistance of M; thus, the pitch of the RNP helix is variable. M distributed on the cytoplasmic side of the envelope interacts with the RNP helix during the budding without securing neighboring turns, resulting in a variable pitch in the RNP helix.

## Data Availability

The density maps of subtomogram averaging (EMD-46612) and single-particle reconstruction (EMD-46621) of RABV have been deposited in the EMDB.

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
