# Peer review of "Structural Heterogeneity of the Rabies Virus Virion"

_viruses, 2024, doi:10.3390/v16091447_

Round 1

Reviewer 1 Report

Comments and Suggestions for Authors

The research of RABV has attracted much attention, not only because of its high pathogenicity, but also its application potential as tool virus in anti-cancer agent or vaccine development. However, compared to VSV, another model rhabdovirus, little is known about the structure and assembly of intact virions of RABV due to difficulties in sample preparation and virion heterogeneity. This work reported the first structural information of assembly of the wild-type RABV virion by combination of cryo-ET and cryo-EM. The main findings include flexibility of the viral morphology, variable pitch and larger diameter range compared to VSV, organizations of M-RNP based on SPA and STA density maps, and prefusion and postfusion G trimers on the viral surface. Although resolutions are limited, the highlights of this paper are obvious: assembly analysis of RABV M and RNP, especially the comparisons between M-N interactions of RABV and VSV, and correlate virus assembly with virus heterogeneity. It is reported that only one layer of M resides in the RABV virion between RNP and envelope, and many interactions of M-M and M-N are lacking, which are quite different from that of VSV who has two layers of M. The authors suppose that the looser organization pattern of RABV gives more structural flexibility, which may help the virus overcome selective pressure.

Overall, the article is well organized and clearly written, and expands the knowledges on the structure of Rhabdoviridae, even on negative-strand RNA viruses. However, in order to be suitable for publication, the authors will need to address the following points:

1.     Lines 234-236: It seems a little contradictory of this sentence with what the authors mentioned earlier in the article: “Continuous iodixanol density gradient ultracentrifugation process was designed to minimize the presence of defective interfering particles with incomplete genomic RNA in RABV virions” (line 194-195). If it’s true, could the authors give explanations about why G gene-deficient RABV do not have defective genomes that influence the size distribution? Otherwise, the reasons of different sizes of virus between the two articles could be different strains used, different data sizes or just normal error range.

2.     Please summarize the cryo-ET/cryo-EM data collection and reconstruction statistics in a supplementary table.

3.     In the RABV trunk model which the author proposed in figure5, polymerase (L) is mentioned but is rarely described in this paper. It’s difficult to identify this protein in the tomogram slices currently provided in the manuscript, could the authors provide a tomogram slice that can identify the protein, and, if possible, give more detailed analyses of this protein in the manuscript, for example, characteristics of quantity and distribution of L on intact virion?

4.     Lines 210-214, figure 1C: Is the alteration in diameter at different height of trunk because of the long axis of virus trunk is not strictly parallel to the ice layer, but tilted? The slice in bottom right of figure 1C is not clear enough to exclude the possibility of the above concern. Could the authors provide a clearer image of virus cross-section, or add some explanations?

5.     Is the virus propagated and purified in a BLS-2 laboratory? Please add more information in Materials and Methods.

6.     Add a figure in supplementary file to describe the workflow of sub tomogram averaging.

7.     Lines 231-234: When compared with the previously reported G gene-deficient RABV, for a better contrast, please also include the length distribution and average length of that G gene-deficient RABV.

8.     Line 360: please provide the PDB ID of RABV N.

Comments on the Quality of English Language

1. Lines 458-461: The description is confusing and meaning of those sentences are repeated. Please simplify and clarify them.

2. There are several errors and omission in the article, please check and correct:

(1)  Line 20: “nucleocapsid protein (N)” should be “nucleoprotein (N)”.

(2)  Line 27: delete “and”.

(3)  Line 32: “ribbit” should be “rabbits’s”.

(4)  Line 84: “Brucket” should be “Bucket”.

(5)  Line 102: “300KV” to “300 kV”.

(6)  Line 138: The author has stated that totally 35 virions were collected as cryo-ET data (Line 132). Please check the sentence “The initial reference map of M-N was generated from ~500 manually picked virions which have the best visibility”.

(7)  Line 323: “the RABV trunk” should be “the RABV M-RNP”.

(8)  Line 364: Only figure H has 8-asymmetry unit, while figure G has two. Please correct.

(9)  Line 406: Here “trunk” is more precise than “entire virion”.

(10) Line 426: “points” to “point”.

 (11) Line 576: Ref33: please supplement title of the article in the citation.

 (12) Supplementary figure 6: In the last sentence of figure legend, please correct “FRC” to “FSC”.

Reviewer 2 Report

Comments and Suggestions for Authors

Despite the fact that three-dimensional structures of structural proteins of the rabies virus have been available for a long time, it was absolutely unclear how these proteins are organized and assembled in the bullet-shaped RABV.

Combining the high resolution data of CryoEM and CryoET approaches the authors now determined the 3D structure of the trunk part of RABV and depicted the architectural organization of the entire bullet-shaped virion. Virions of various morphology were described. The RABV's architecture was compared to the architecture of its close relative - the VSV's virion resolved earlier.

Importantly, it was demonstrated that RABV has a single layer of M protein molecules that binds with the underlying N-RNA ribonucleoprotein helix differently as compared to VSV, which has two layers of M securing the N bullet. 

In addition, conformations of pre- and post-fusion conformations of G-protein homotrimers were reliably detected on the surface of rabies virus virions, that may be important in future for characterizing the quality of potential vaccine and anticancer preparations obtained on the basis of recombinant rabies virus.

The paper is of high quality both in terms of scientific content and design, contains original structural data that have not been previously published. Thus, in principle, it warrants publication as it is. Nevertheless, I would like to suggest some small discussion regarding the stoichiometry of the M and N proteins within the RABV and VSV virions, which the authors postulate in the present and their previous paper [doi: 10.1038/s41467-022-33664-4] based on the Cryo EM/ET data. I wonder if these ratios match the biochemical data? Often the structural protein ratios within the virions influence both the morphology and the life cycle of the virus. I would advise the authors (probably in the Supplemental Material or in the main body of the Ms.) to provide a comparative image of the Laemmli electrophoretic separation of the proteins of RABV and VSV virions, demonstrating the ratio M:N equal to 1:1 and 2:1, respectively, according to the stated, or refer to the relevant publications.

It would also be very interesting to receive some comments from the authors as a discussion regarding the CryoEM data obtained earlier for filamentous influenza virions [e.g. doi: 10.1073/pnas.1002123107]. Despite the influenza virions have the M1:N ratios that may differ greately depending on the virus strain, it is more often around 2:1 or even more [doi: 10.1006/viro.1997.8916]. At the same time all the CryoEM/ET images and 3D-reconstructions done so far show only one layer of M1 protein molecules under the influenza viral envelope. This is probably (one of) the reasons that influenza virions are not as rigid and homogeneous in morphology as VSV and to a lesser extent RABV, on the contrary, they are very pleomorphic.

As a minor point:

Line 32: is it rabbit (not ribbit) meant?

Reviewer 3 Report

Comments and Suggestions for Authors

This is a study that builds on many in the past. The sturcture of bovine ephemeral fever challenged us in the 1970s. It is great to see these contemporary techniques being applied.

Comments on the Quality of English Language

Minor changes

Line 87 should this be pooled.

Line 98 and 112 is it grid or grids?
